# Ocular Formulation Based on Palmitoylethanolamide-Loaded Nanostructured Lipid Carriers: Technological and Pharmacological Profile

**DOI:** 10.3390/nano10020287

**Published:** 2020-02-08

**Authors:** Carmelo Puglia, Debora Santonocito, Carmine Ostacolo, Eduardo Maria Sommella, Pietro Campiglia, Claudia Carbone, Filippo Drago, Rosario Pignatello, Claudio Bucolo

**Affiliations:** 1Department of Drug Sciences, University of Catania, Viale Andrea Doria 6, 95125 Catania, Italy; debora.santonocito@outlook.it (D.S.); ccarbone@unict.it (C.C.); r.pignatello@unict.it (R.P.); 2Department of Pharmacy, University Federico II of Naples, Via D. Montesano 49, 80131 Naples, Italy; ostacolo@unina.it; 3Department of Pharmacy, University of Salerno, Via G. Paolo II, 84084 Fisciano (SA), Italy; esommella@unisa.it (E.M.S.); pcampiglia@unisa.it (P.C.); 4Department of Biomedical and Biotechnological Sciences, University of Catania, Via Santa Sofia 97, 95123 Catania, Italy; f.drago@unict.it (F.D.); bucocla@unict.it (C.B.)

**Keywords:** palmitoylethanolamide, nanostructured lipid carriers, ocular drug delivery, Turbiscan technology, diabetic retinopathy

## Abstract

The present work was aimed for the preparation of a stable nanostructured lipid carrier (NLC) system for the delivery of N-palmitoylethanolamide (PEA) to the back of the eye. PEA is an interesting natural compound showing anti-inflammatory and neuroprotective activities. The limits of PEA (poor solubility and high instability) justify its nanoencapsulation into drug delivery systems. Two different well-known techniques were compared to formulate NLC: the high shear homogenization technique (HSH) and the method based on a combination of HSH technique and ultrasonication (HSH/US). Nanoparticles were evaluated in relation to mean size, homogeneity, surface charge, and physical stability by Turbiscan technology. Retinal distribution of PEA was carried out in a rat eye after single instillation of PEA-NLC ophthalmic formulation. The novel formulation delivered remarkable levels of PEA to the retina. Lastly, topical administration of PEA-NLC ophthalmic formulation was able to significantly inhibits retinal tumor necrosis factor-α (TNF-α) levels in streptozotocin-induced diabetic rats. The present findings suggest that the novel ophthalmic formulation may be useful for the treatment of retinal diseases such as diabetic retinopathy. Clinical studies are in progress to evaluate this possibility.

## 1. Introduction

Diabetic retinopathy (DR) is likely the most frequent cause of loss of sight and visual impairment and it is strongly influenced by many factors such as diabetes duration, poor glycemic control, and hypertension [1]. In diabetes, sustained hyperglycemia works as a trigger for a variety of events leading in vascular dysfunction. Vascular alterations in DR are related to several biochemical and immunological mechanisms that promote pro-inflammatory cytokines release such as TNF-α. Several studies showed increased levels of retinal TNF-α in diabetic patients and animals as well [2,3,4,5,6,7]. The inflammation process comprises a variety of functional and molecular mediators, including recruitment and/or activation of leukocytes, while it is well known that the expression of several inflammatory biomarkers is regulated by the activation of proinflammatory transcription factors such as Nuclear Factor-kappa-B (NF-kB). Activation of NF-kB leads to the modulation of many cytokines such as TNF-α [8]. N-palmitoylethanolamide (PEA) is an endogenous fatty acid amide belonging to the family of N-acylethanolamines (NAEs). Several works outlined an anti-inflammatory and neuroprotective action of PEA that can be useful in the treatment of different pathological conditions such as retinal diseases [9,10,11]. The pharmacological features of PEA involve effects upon mast cells, cannabinoid receptors, potassium channels, transient receptor potential channels, NF-kB, and the nuclear receptor peroxisome proliferator-activated receptor α (PPARα) [12,13]. Several studies demonstrated the beneficial role of PEA in retinal diseases such as DR, which preserves the integrity of the blood–retinal barrier [9,14,15]. 

Despite these important effects, the clinical use of PEA is limited by an unfavorable pharmacokinetic profile due to the scarce water solubility and the lack of stability that limit the development of conventional eye drops [16]. To overcome these critical issues, a nanostructured lipid carriers (NLC) approach has been explored. These new nanocarriers demonstrated the possession of important beneficial features for ocular application such as controlled drug release, high drug encapsulation efficiency, and a noteworthy biocompatibility and tolerability [14]. On the basis of previous considerations, the present study was aimed to investigate the novel PEA-NLC formulations in terms of retinal distribution and in terms of pharmacological effects in a normal rat eye and a diabetic rat eye, respectively. Furthermore, we provided additional features regarding the long-term stability of formulation and the high potential of these nanocarriers as vehicles for drug ocular delivery.

## 2. Materials and Methods 

### 2.1. Materials

Compritol 888 ATO (COMP), a mixture of monoglycerides, diglycerides, and triglycerides of behenic acid was obtained from Gattefossè (Milan, Italy). Miglyol 812 (MIG), which is a mixture of caprylic/capric triglycerides, was obtained from Eigenmann & Veronelli S.p.A. (Milan, Italy) and Lutrol F68 was provided by BASF ChemTrade GmbH (Burgbernheim, Germany). Micronized PEA was a kind gift from Epitech Group (Milan, Italy). Streptozotocin (STZ) was purchased from Sigma-Aldrich (St. Louis, MO, USA). All the other chemicals and reagents were of the highest purity grade commercially available and were used as received.

### 2.2. Preparation of PEA Formulations

The formulation of the PEA-NLC was preceded by a pre-formulation study, which was necessary to optimize the technological parameters of the NLC. PEA-NLCs were generated using a method previously described [17], with some modifications, by high shear homogenization coupled to ultrasound (HSH-US). A total of 0.2 g of PEA were dissolved in an oil phase containing MIG (0.4 g) and COMP (0.6 g) and the mixture was stirred at 80 °C in order to get a dispersion. The aqueous phase consisted of Lutrol (0.1 g) and distilled water (25 mL). The melted lipid phase was dispersed in the hot (80 °C) aqueous phase by using a high-speed stirrer (Ultra-Turrax T25, IKA-Werke GmbH &Co. Kg, Staufen, Germany) at 13,500 rpm for 10 min, which maintains the temperature at least 10 °C above the lipid melting point. The obtained pre-emulsion was ultra-sonified by a Labsonic 2000 (B. Braun, Melsunen, Germany) for 8 min and then cooled by dilution in 25 mL of water at 4 °C. The same procedure was followed to formulate the unloaded NLC without adding the drug. A PEA suspension has been formulated as reference formulation for an in vivo study, by the addition, under stirring, of 0.2 g of PEA to a weighted amount of distilled water. In all the formulations, the PEA final concentration was 0.4% (w/v).

### 2.3. PEA-NLC Characterization

The average size (Z-Ave) and polydispersity index (PDI) of the nanoparticles were measured by photon correlation spectroscopy (PCS). A Zeta Sizer Nano-ZS90 (Malvern Instrument Ltd., Worcs, England), equipped with a solid-state laser having a nominal power of 4.5 mW with a maximum output of 5 mW 670 nm, was employed. Analyses were performed using a 90° scattering angle at 20 ± 0.2 °C. Samples were prepared diluting 100 μL of NLC suspension with 900 μL of distilled water. The Zeta Potential (ZP, ξ) is an indicator of the stability of a dispersed system. The instrument used for the measurement gave us indications about the electrophoretic mobility of particles in dispersion or in solution. The instrument carried out three sets of measures up to 100.0 to achieve an average value.

### 2.4. Determination of PEA Loading

The percentage of PEA entrapped in the lipid matrix was determined as follows: NLC dispersions (NLC_1_ and NLC_2_) were filtered by using a Pellicon XL tangential ultrafiltration system (Millipore, Milan, Italy) equipped with a polyethersulfone Biomax 10 membrane. Some amount of retained material was freeze-dried, dissolved in chloroform, and analyzed by the UHPLC method described below. PEA incorporation efficiency was expressed both as encapsulation efficiency (E.E.%) and drug loading (D.L.%), calculated from Equations (1) and (2), respectively.
E.E. (%) = (Mass of PEA in nanoparticles)/(Mass of PEA fed to the system)(1)
D.L. (%) = (Mass of PEA in nanoparticles)/(Mass of nanoparticles)(2)

### 2.5. Stability Studies

NLC stability was evaluated by Turbiscan® Ageing Station (TAGS) (Formulaction, l’Union, France). Twenty milliliters of each NLC were placed in a cylindrical glass cell and positioned in the Turbiscan® at 25 or 35.5 ± 2 °C for 14 days. The detection head was composed of a pulsed near-infrared light source (λ = 880 nm). Two synchronous transmission (T) and back scattering (BS) detectors receive the light that crosses the sample (at 180° from the incident beam) for the T detector, while the BS detector takes the light scattered backwards by the sample (at 45° from the incident beam). The detection head scanned the entire height of the sample cell (65 mm in longitude), acquiring T and BS each of 40 mm (1625 acquisitions in each scan). The Turbiscan® makes scans at various pre-programmed times and overlays the profiles on one graph in order to show possible destabilization phenomena. In our experiments, the stability of the samples was evaluated based on the variation of backscattering (ΔBS), which is shown in ordinate, while the height of the cell is reported in abscissa.

### 2.6. In Vivo Study

#### 2.6.1. Animals

Rats (Male Sprague-Dawley) weighing 175–200 g (Envigo, San Pietro a Nadisone, Udine, Italy) were used. Animals were housed in a light and temperature-controlled room with tap water and standard chow provided ad libitum. Experimental animal procedures followed guidelines of the Animal Care and Use Committee of the University of Catania and conformed to the Association for Research in Vision and Ophthalmology (ARVO) resolution on the use of animals in research. Final group sizes for all measurements were *n* = 6–9 except as noted. STZ was obtained from Sigma-Aldrich (Milan, Italy).

#### 2.6.2. Induction of Diabetes

Streptozotocin (STZ)-induced diabetes has been widely used as an animal model for type 1 diabetes mellitus studies [18]. STZ-induced diabetic rats demonstrate characteristics of non-proliferative diabetic retinopathy seen in humans including inflammatory mediators release. After 12 h of fasting, the animals received a single 60 mg/kg intravenous (i.v.) injection of STZ in 10 mM sodium citrate buffer, pH 4.5 (1 mL/kg dose volume). Control (sham, non-diabetic) animals were fasted and received citrate buffer alone. After 24 h, animals with blood glucose levels >250 mg/dl were considered diabetic. The diabetic state was assessed by a blood glucose meter (Accu-Check Active1, Roche Diagnostic, Milan, Italy). Topical administration of PEA-NLC formulation (10 µL/eye, TID for 10 days) was instilled in the conjunctival sac. All the experiments were performed 10 days following the induction of diabetes.

#### 2.6.3. TNF-α Assessment

Rat eyes (*n* = 12 per group) were collected 10 days after STZ administration, and each retina was homogenized in 100 mL of digest solution as previously described [19]. The solution was supplemented with a cocktail of protease inhibitors (Complete Protease Inhibitor Cocktail, Roche, Basel, Switzerland) before use. Samples were spinned (10 min at 10,000× g) and assessed for protein concentration with the bicinchoninic acid (BCA) assay (Mini BCA Kit, Thermo Fisher Scientific, Waltham, MA, USA). The TNF-α levels were evaluated by ELISA (R&D Systems, Minneapolis, MN), according to the manufacturer’s instructions. All measurements were performed in duplicate. The tissue sample concentration was calculated from a standard curve and corrected for protein concentration.

#### 2.6.4. Pharmacokinetics Study

Rats were treated with a single ocular topical administration (10 µL) of PEA-NLC formulation. At predetermined intervals (30’, 60’, 90’ and 240’) after ocular administration of PEA-NLC formulation the rats were sacrificed and their eyes were enucleated. The eyes were then cut from the equator, removing the lens and vitreous. The retina was carefully dissected from the choroid and optic nerve. Retinal samples were stored at −80 °C until analysis. Pharmacokinetic studies took into account the following parameters: a) peak eye tissue concentration (Cmax); b) time of peak of eye tissue concentration (Tmax); c) area under the curve (AUC) of PEA tissue concentration [PEA] vs. time curve from 0 to 240’(AUC0-240). Tissue levels of PEA were normalized to PEA content in control animals. All results were reported as mean ± SD.

#### 2.6.5. Ocular Tolerability

In a separate set of animals, we carried out the ocular tolerability. Briefly, topical administration of PEA-NLC formulation (10 µL/eye, TID for 10 days) was performed and all eyes were evaluated by slit lamp at day 1, 5, and 10. The severity of conjunctival hyperemia scoring (none = 0, mild = 1, moderate = 2, severe = 3) was performed by an investigator unaware of the treatment.

### 2.7. Analytical Methods

#### 2.7.1. Sample Preparation

Frozen tissues were first lyophilized (LyoQuest-55, Telstar Technologies, Spain) for 24 h. One mg of the lyophilized was extracted with 300 μL of ice-cold acetonitrile containing 20 ng/mL of deuterated PEA (d-PEA). Samples were vortexed for 20 s and then centrifuged for 5 min at 16,000 rpm (Eppendorf). The supernatant was filtered on a PTFE membrane (0.22 μm, Phenex, Phenomenex, Castel Maggiore, BO, Italy) and injected in the UHPLC system.

#### 2.7.2. Instrumentation

UHPLC-MS/MS analysis was carried out with a Shimadzu Nexera (Shimadzu, Milan, Italy) UHPLC consisting of two LC 30 AD pumps, a SIL 30 AC autosampler, a CTO 20 AC column oven, a CBM 20A controller, and the system was coupled online to a triple quadrupole LCMS 8050 (Shimadzu, Kyoto, Japan) by an electrospray ionization source.

#### 2.7.3. UHPLC-MS/MS Conditions

The separation was performed on an Ascentis Express® HILIC column with geometry (L × I.D) 100 × 2.1 mm, 2.7 μm (Supelco, Bellefonte, PA, USA) employed as mobile phases: A) water containing 50 mM ammonium formate and B) acetonitrile (ACN) with a gradient starting from 99% B, 0.01–2.50 min, 85% B, 2.51–3.00 min, 85%–40% B, then isocratic for 1.00 min, and returning to 99% B in 6 min. The flow rate was set at 0.5 mL/min. The column oven was set at 35 °C and 2 μL of extract were injected. All additives and mobile phases were of an LC-MS grade and purchased from Sigma Aldrich (Milan, Italy). The ESI was operated in a positive ionization. MS/MS analysis of PEA were conducted in multiple reaction monitoring (MRM), employing as transitions: 300.10 > 62.10 (quantifier ion), Q1 pre bias −16.0 V, collision energy: −14.0 V, Q3 pre bias −24.0 V, 300.10 > 57.15 (qualifier ion) Q1 pre bias −15.0 V, collision energy: −27.0 V, Q3 pre bias −23.0 V. Dwell time was 10 ms. In addition, MS/MS analysis of the internal standard deuterated-PEA were conducted in multiple reaction monitoring (MRM), employing as transitions: 304.10 > 62.15 (quantifier ion), Q1 pre bias −15.0 V, collision energy: −16.0 V, Q3 pre bias −25.0 V; 304.10 > 58.10 (qualifier ion), Q1 pre bias −15.0 V, collision energy: −28.0 V, Q3 pre bias −23.0 V, and dwell time of 10 ms. Interface temperature, desolvation line temperature, and heat block temperature were set, respectively, at 300 °C, 250 °C, and 400 °C. Nebulizing gas and drying (N_2_) and heating gas (air) were set, respectively, at 3, 10, and 10 L/min. PEA stock solution (1 mg/mL) was prepared in ACN and the calibration curve was obtained in a concentration range of 0.5–50 ng/mL (R2 = 0.999) and Deuterated-PEA (Cayman Chemicals) was selected as an internal standard in the concentration of 20 ng/mL. Repeatability was established by duplicate injections of sample and solutions at low, medium, and high concentration levels of the calibration curve with the same chromatographic conditions and analyst at the same day and within two consecutive days. Repeatability was established by duplicate injections of sample and solutions at low, medium, and high concentration levels of the calibration curve with the same chromatographic conditions and analyst at the same day, which shows good retention time and ng/mg extract repeatability with maximum CV% values ≤ 0.15 and 4.51, respectively. Limits of detection (LODs) and quantification (LOQs) were calculated by the ratio between the standard deviation (SD) and analytical curve slope multiplied by 3 and 10, respectively, obtaining as values: LOQ = 0.05 ng/mL, LOD = 0.001 ng/mL.

### 2.8. Statistical Analysis

All values are expressed as mean +/− SD. The results were analyzed by one-way ANOVA, which was followed by the Dunnett post hoc test. Differences between groups were considered significant given p-values < 0.05.

## 3. Results and Discussion

PEA-NLC morphological evaluation has been investigated in a previous work of our group [11,14]. FE-SEM analyses, in particular, showed in the sample low-electron density spherical objects likely related to the occurrence of Miglyol® 812 oily nano-compartments dispersed in Compritol® 888 ATO solid matrix, as previously hypothesized [20]. This peculiar internal structure of the NLC is recognized as “multiple type” and is typical of nano-dispersions characterized by a relative high oil content and a reduced miscibility during particle solidification. The last feature, in particular, causes the separation of the oil, which leads to the formation of nano-compartments within the solid matrix. 

In the present work, NLC have been formulated using two different preparation strategies with the final aim to identify the best conditions to obtain a product endowed with high stability over time. Therefore, we evaluated two well-known techniques used to formulate lipid nanoparticles such as the high shear homogenization technique (HSH) and the method based on a combination of the HSH technique and ultrasonication (HSH/US) [17,21]. The formulations generated (NLC_1_ for HSH/US and NLC_2_ for HSH techniques, respectively) were fully characterized in order to obtain information about the nanoparticle characteristics. PCS data proved that NLC_1_ provided better results in terms of mean diameters (Z-Ave), population homogeneity (PDI), ZP, percentage of encapsulation efficiency (E.E.%), and percentage of drug loading (D.L.%) compared to NLC_2_ formulation (Table 1). 

These results are in accordance with the findings reported in scientific literature regarding the important role of US in controlling the final dimension of lipid nanoparticle dispersions [22]. Therefore, the method based on the combination of the HSH technique and ultrasonication (HSH/US) can guarantee a valid nanoparticle size and a good population homogeneity. In fact, the dispersion quality obtained by high shear homogenization is often compromised by the presence of microparticles, which can significantly affect the Z-Ave determination [23]. Furthermore, the HSH/US method used to formulate NLC_1_, was useful to increase the E.E.% (from 20.6% to 82.3%) and D.L.% (from 0.08% to 0.32%) in comparison with NLC_2_, which is formulated by the HSH technique (Table 1). 

In order to evaluate the stability of the prepared NLC_1_ and NLC_2_, we used Turbiscan®, which is a well-known technology described in literature to achieve objective information about the physical stability of colloidal suspensions in terms of particle migration or aggregation [24,25,26,27,28,29,30,31]. As clearly shown in the destabilization kinetics reported in Figure 1, NLC_1_ is very stable at both 25 and 35.5 °C, since no significant variation in the Turbiscan® Stability Index (TSI) was shown after 14 days of storage. However, the occurrence of instability phenomena was observed in sample NLC_2_, even after 1 day of storage.

In order to further investigate the instability phenomena, we evaluated the backscattering profiles of the formulations stored at 25 °C and 35.5 °C (Figure 2 and Figure 3). Furthermore, we observed a linear profile for NLC_1_, related to high stability at both temperatures, that clearly reveals the absence of particle aggregation and only a slight particle migration toward the bottom of the cuvette, which can be considered insignificant (ΔBS<<10%). On the other hand, the backscattering profiles of NLC_2_ highlight the occurrence of important instability phenomena at both the tested temperatures, which are mainly related to particle aggregation, as clearly shown by the ΔBS values ≥ 20%.

Since NLC_1_ was prepared using the HSH/US method, while Ultraturrax only was used to obtain NLC_2_. It is possible to hypothesize that the higher energy applied in the emulsification of lipids in the aqueous phase could be responsible for the formation of smaller particles with greater homogeneity, which consequently affects the sample’s physical stability, as confirmed by Turbiscan® analysis. These results are in line with previous findings, which demonstrates the greater physical stability of NLC prepared with high pressure homogenization (HPH) compared to those obtained by the phase inversion temperature method (PIT) [31]. In particular, sonication can be used not only to reduce the mean size of preformed lipid nanoparticles [32], but also as a key step able to affect the interaction between particles in lipid and aqueous phases, which can be exploited as an intermediate step for enabling the emulsification of lipids in the aqueous phase. This was followed by a proper preparation method [33].

PEA-NLC ophthalmic formulation was assessed in vivo in order to investigate the retinal distribution profile of PEA and to figure out the pharmacological effect of the drug in a DR model.

TNF-α is an inflammatory cytokine involved in the ocular inflammatory process and it is the main cause of the blood-retinal barrier breakdown (BRB) during DR. The induction of TNF-α is most likely due to some mediators that operate during the course of diabetes, such as NF-κB. The present study demonstrated that PEA once loaded in NLC was able to reduce the amount of TNF-α in retinal tissues collected 10 days after STZ injection (Figure 4). This is likely due to the fact that PEA does not reach the back of the eye when administered as suspension (see below). 

These results confirm the evidence observed by Bucolo and co-workers [34] regarding the remarkable increase of retinal TNF-α in STZ-induced diabetic rats and by Paterniti and coworkers about the protective effect of PEA on DR [15], and emphasize the importance of a drug targeting strategy in order to maximize PEA pharmacological activity. The pharmacokinetic profile of a single dose (10 µL/eye) of PEA-NLC formulation was compared to the same dose of PEA aqueous suspension (PEA suspension). When administered as suspension, PEA is not able to reach the rat retina, while, after encapsulation in lipid nanoparticles (PEA-NLC), the drug reached detectable levels in this tissue (Cmax = 8.3 ng/g) (Figure 5).

Furthermore, we assessed the ocular tolerability of the formulation in the rat eye and a high safety profile has been observed, in particular the conjunctival hyperemia score was zero at all time (data not shown). These data are in accordance with our previous study on the rabbit eye [14]. We previously demonstrated that, after instillation of PEA-NLC formulation in the rabbit eye, PEA moved from the anterior to the posterior segment of the eye following a non-corneal route. The mechanism likely involved a conjunctival/scleral absorption followed by distribution to choroid, vitreous, and retina, even though it is very difficult to rule out a classical corneal absorption.

## 4. Conclusions

NLC_1_ formulated following a method based on a combination of the HSH technique and ultrasonication (HSH/US) provided better results in terms of mean particle size (Z-Ave), population homogeneity (PDI), and ZP with respect to NLC_2_ formulated by a high shear homogenization technique (HSH). This result outlines the important role of US in controlling the final dimension of a lipid nanoparticle dispersion. Stability studies by Turbiscan® AGS demonstrated that NLC_1_ have a greater physical stability compared to NLC_2_ due to the higher energy applied in the emulsification process used in the HSH/US method. We demonstrated that sonication represents a key intermediate step for the formation of colloidal suspensions of small homogeneously dispersed nanoparticles with very high stability without the occurrence of particle migration/aggregation phenomena during storage. Pharmacokinetics data showed that PEA was able to reach the back of the eye once enclosed in NLC. Therefore, the results of the present study demonstrate that PEA-NLC-based formulation for ophthalmic application, has a unique pharmacological profile both in terms of retinal distribution and inhibition of inflammation. Therefore, these findings suggest that the novel ophthalmic formulation may be useful in the management of retinal diseases such as diabetic retinopathy. Clinical studies are planning to explore this possibility.

## Figures and Tables

**Figure 1 nanomaterials-10-00287-f001:**
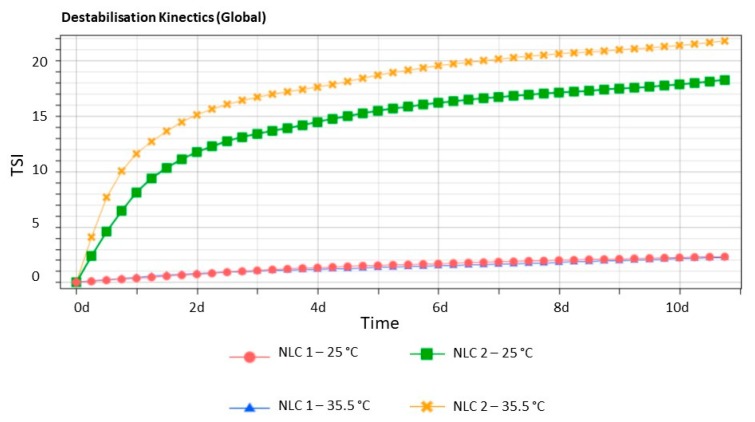
Destabilization kinetics represented in terms of evolution of Turbiscan® Stability Index (TSI) of NLC_1_ and NLC_2_ stored 14 days at 25 and 35.5 °C.

**Figure 2 nanomaterials-10-00287-f002:**
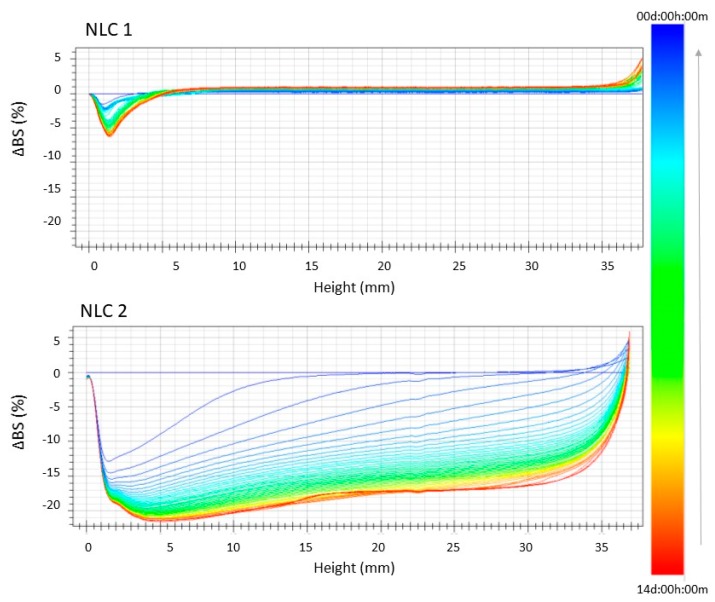
Backscattering profiles (ΔBS) of NLC_1_ and NLC_2_ stored in Turbiscan® at 25.0 ± 1.0 °C. Data are represented as a function of time (0–14 days) of sample height (0–20 mm). The sense of analysis time is indicated by the arrow.

**Figure 3 nanomaterials-10-00287-f003:**
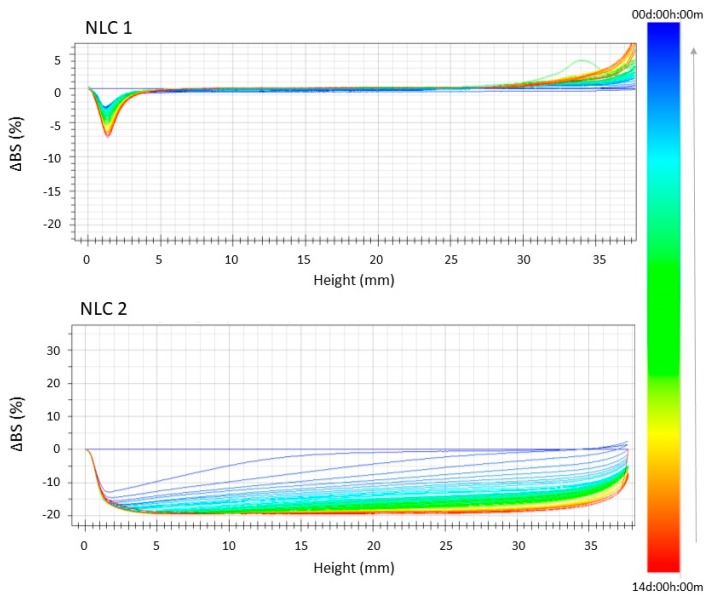
Backscattering profiles (ΔBS) of NLC_1_ and NLC_2_ stored in Turbiscan® at 35.5 ± 1.0 °C. Data are represented as a function of time (0–14 days) of sample height (0–20 mm). The sense of analysis time is indicated by the arrow.

**Figure 4 nanomaterials-10-00287-f004:**
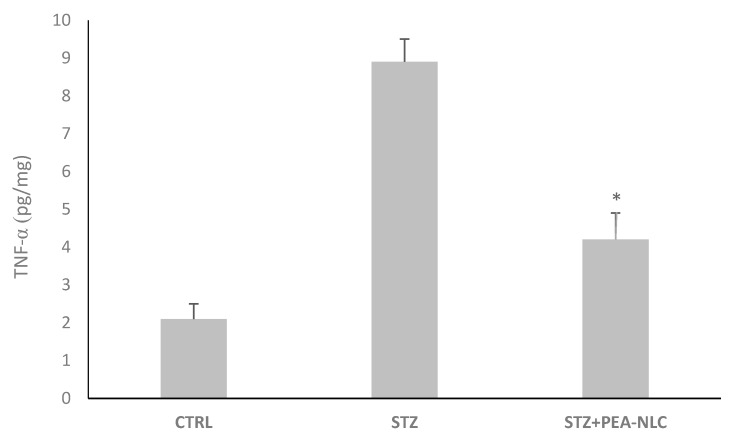
Retinal levels of TNF-α 10 days after STZ injection. Data shown are expressed as the mean ± SD. **p* < 0.05 vs. STZ. *n* = 12.

**Figure 5 nanomaterials-10-00287-f005:**
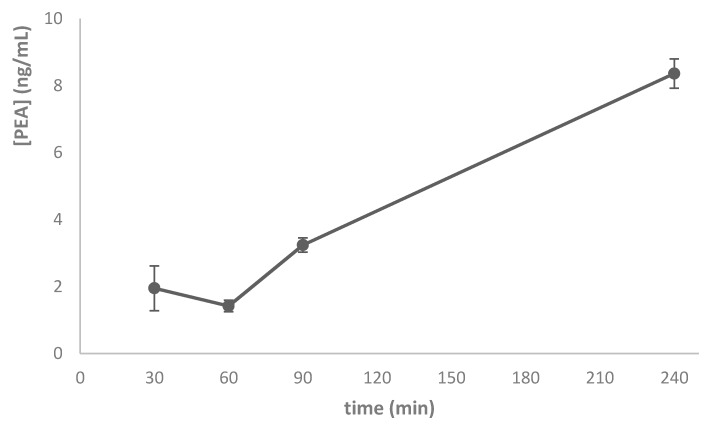
Retinal PEA distribution after instillation of a single dose of PEA-NLC formulation.

**Table 1 nanomaterials-10-00287-t001:** Mean particle size (Z-Ave), polydispersity index (PDI), zeta potential (ZP), encapsulation efficiency (E.E. %), and drug loading (D.L. %) of NLC_1_ and NLC_2_. Values are the mean of at least three measures.

Form	Z-Ave ± S.D. (nm)	PDI ± S.D.	ZP ± S.D. (mV)	E.E. (%) ± S.D.	D.L. (%) ± S.D.
**NLC_1_**	264.5 ± 0.19	0.200 ± 0.035	−37.1 ± 0.02	82.3 ± 0.82	0.32 ± 0.02
**NLC_2_**	731.2 ± 0.21	0.269 ± 0.038	−41.4 ± 0.01	20.6 ± 0.76	0.08 ± 0.01

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
