# Peer review of "Ocular Formulation Based on Palmitoylethanolamide-Loaded Nanostructured Lipid Carriers: Technological and Pharmacological Profile"

_nanomaterials, 2020, doi:10.3390/nano10020287_

Round 1
Reviewer 1 Report
This is well-conceived and well executed work. The presentation of the stability measurements is convincing; Figures 1, 2, 3 all show NLC1 is quite stable while NLC2 is unstable.
Minor points
Line 78 Was the temperature of the ultrasonifier procedure also 10C above the lipid melting point, as mentioned in the previous sentence?
Line 165, line 179 missing units
Line 245 Small English syntax problem; believe the intent was, “In order to further investigate the instability …..”
Author Response
Minor points
- Line 78 Was the temperature of the ultrasonifier procedure also 10C above the lipid melting point, as mentioned in the previous sentence?
Answer: The temperature was always around 80°C during all the steps of the experiment.
- Line 165, line 179 missing units
Answer: The typos have been corrected.
- Line 245 Small English syntax problem; believe the intent was, “In order to further investigate the instability …..
Answer: Thank you. Now it sounds better!
Reviewer 2 Report
This paper reports the preparation of PEA-NLC nanoparticles and the inhibition of retinal TNF-a levels in STZ-induced diabetic rats. Two methods for the preparation of PEA-NLC nanoparticles are reported. The first method, which makes use of the HSH method and leads to NLC2, has already been published [Ref. 11, 14]. The second method, which makes use of the HSH/US and leads to NLC1, gave nanoparticles with good physicochemical characteristics (relatively good size and pdi, high encapsulation efficiency and drug loading).
One question is why in this work NLC2 nanoparticles were found to have such a high size, while in Ref. 11 and 14, the same method, gave PEA-NLC nanoparticles with size around 150-200nm.
My second question is why the authors claim that the morphological evaluation of NLC1 has already been done in previous work, referring to publications 11, 14, while in these works only the HSH method was used for the preparation of the PEA-NLC nanoparticles. In my opinion, PEA-NLC1 (prepared by HSH/US method) should be examined by TEM/FE-SEM for their morphological characteristics, as well.
Regarding the experiment for the inhibition of THF-a levels, this is promising and of interest, and deserves publication. One question is what are the levels of TNF-a (Figure 2) by the application of PEA alone?
Regarding Figure 5 and the claim that PEA suspension is not able to reach the rat retina, this is not obvious in figure 5 (as mentioned in lines 288, 289). You should better remove “(figure 5)” from lines 288,289 and put it in the end of the sentence (line 290). Also, it should be mentioned in the caption of Figure 5 that the curve refers to PEA accumulation after the application of PEA-NLC nanoparticles.
Regarding the tolerability of the formulations, further discussion is necessary. What is the score?
Author Response
- One question is why in this work NLC2 nanoparticles were found to have such a high size, while in Ref. 11 and 14, the same method, gave PEA-NLC nanoparticles with size around 150-200nm.
- My second question is why the authors claim that the morphological evaluation of NLC1 has already been done in previous work, referring to publications 11, 14, while in these works only the HSH method was used for the preparation of the PEA-NLC nanoparticles. In my opinion, PEA-NLC1 (prepared by HSH/US method) should be examined by TEM/FE-SEM for their morphological characteristics, as well.
Answers:
Thank you for your right analysis. Clearly an explanation is needed. Since the first paper [Tronino, D.; Offerta, A.; Ostacolo, C.; Russo, R.; De Caro, C.; Calignano, A.; Puglia, C.; Blasi, P. Nanoparticles prolong N-palmitoylethanolamide anti-inflammatoryand analgesic effects in vivo. Colloids Surf B Biointerfaces 2016, 141, 311–317] PEA-loaded NLC were prepared by high shear homogenization and ultrasonication (HSH-US) method as reported by the reference 19 of that manuscript [C. Puglia, A. Offerta, L. Rizza, G. Zingale, F. Bonina, S. Ronsisvalle, Optimization of curcumin loaded lipid nanoparticles formulated using high shear homogenization (HSH) and ultrasonication (US) methods, J. Nanosci. Nanotechnol. 13 (2013) 6888–6893].
Also in the second manuscript [Puglia, C.; Blasi, P.; Ostacolo, C.; Sommella, E.; Bucolo, C., Platania, C.B.M.; Romano, G.L.; Geraci, F., Drago, F.; Santonocito, D.; Albertini, B.; Campiglia, P.; Puglisi, G.; Pignatello, R. Innovative Nanoparticles Enhance N-Palmitoylethanolamide Intraocular Delivery. Front Pharmacol 2018, 9, 285] we used the same technique (HSH/US method).
My error during the elaboration of these manuscripts was the writing of a paragraph with a very brief description about the preparation, giving a marginal role to US method step.
Conversely, the optimization of the experimental protocol has been one of the important aims of this last paper and so we explained the pivotal role of US coupled HSH to obtain nanoparticles with optimal nanotechnological parameters. In order to produce a valid interpretation of results we decided to formulate the NLC2, prepared ONLY by HSH method.
For these reasons the morphological evaluation in the previous papers has been made on NLC formulated by HSH/US method and therefore we consider quite completed the experiments done to study the characteristics of the systems.
Regarding the experiment for the inhibition of THF-a levels, this is promising and of interest, and deserves publication. One question is what are the levels of TNF-a (Figure 2) by the application of PEA alone?
Answer: treatment with PEA alone (suspension) in STZ-rats did not decrease TNF-alpha levels; in fact, the TNF-alpha levels were almost the same of the STZ group (data not shown). This is probably due to the fact that PEA does not reach the back of the eye when administered as suspension.
Regarding Figure 5 and the claim that PEA suspension is not able to reach the rat retina, this is not obvious in figure 5 (as mentioned in lines 288, 289). You should better remove “(figure 5)” from lines 288,289 and put it in the end of the sentence (line 290). Also, it should be mentioned in the caption of Figure 5 that the curve refers to PEA accumulation after the application of PEA-NLC nanoparticles.
Answer: ok, we now fix it, and also added more information in the caption.
Regarding the tolerability of the formulations, further discussion is necessary. What is the score?
Answer: we added more information on page 10
Round 2
Reviewer 2 Report
The authors answered/explained all questions raised by my previous report, also explaining why there is no need for an extra morphological experiment. In my opinion the article can be published.